# *In vivo* elongation of thin filaments results in heart failure

**Lei Mi-Mi**[ID]**, Gerrie P. Farman, Rachel M. Mayfield, Joshua Strom, Miensheng Chu, Christopher T. Pappas, Carol C. Gregorio***

Department of Cellular and Molecular Medicine and Sarver Molecular Cardiovascular Research Program, The University of Arizona, Tucson, AZ, United States of America

* gregorio@email.arizona.edu

## Abstract

A novel cardiac-specific transgenic mouse model was generated to identify the physiological consequences of elongated thin filaments during post-natal development in the heart. Remarkably, increasing the expression levels *in vivo* of just one sarcomeric protein, Lmod2, results in ~10% longer thin filaments (up to 26% longer in some individual sarcomeres) that produce up to 50% less contractile force. Increasing the levels of Lmod2 *in vivo* (Lmod2-TG) also allows us to probe the contribution of Lmod2 in the progression of cardiac myopathy because Lmod2-TG mice present with a unique cardiomyopathy involving enlarged atrial and ventricular lumens, increased heart mass, disorganized myofibrils and eventually, heart failure. Turning off of *Lmod2* transgene expression at postnatal day 3 successfully prevents thin filament elongation, as well as gross morphological and functional disease progression. We show here that *Lmod2* has an essential role in regulating cardiac contractile force and function.

**Data Availability Statement:** The raw blot data underlying this study have been deposited to OSF and may be accessed via DOI: 10.17605/OSF.IO/GMEZ8 (URL: https://osf.io/gmez8/). All other

## Introduction

Contractile force in striated muscles is produced by the concerted interaction between interdigitating actin-based thin filaments and myosin-based thick filaments. As such, precise maintenance of filament lengths is essential for efficient contraction. Although it is known that changes in thin filament lengths are linked to development of cardiac and skeletal myopathies [1–5], how those changes contribute to the pathophysiological mechanism of disease progression has yet to be shown. Numerous actin-binding proteins have been shown to regulate the lengths of actin filaments from their barbed ends in non-muscle cells; however, in mammalian cardiac muscle cells, where dynamic regulation of thin filament lengths occurs from the pointed ends in the center of the sarcomere, tropomodulin and leiomodin are the only proteins reported to localize to the pointed ends, and function to maintain thin filament lengths [reviewed in [6–8]].

Leiomodin (Lmod) and tropomodulin (Tmod) proteins are structurally similar to each other, with Lmods containing a C-terminal extension with an extra actin-binding site [9–14]. Each of three leiomodin isoforms show a predominant expression pattern in different muscle

relevant data are within the paper and its
Supporting Information files.

**Funding:** This work was supported by American
Heart Association Postdoctoral Fellowship Award
#16POST31350016, and a Sarver Heart Center
Novel Research Project Award (Frank and Alex
Frazer, and Mark Schiffman) in the Area of
Cardiovascular Disease and Medicine to L.M.M.,
and National Institutes of Health #R01HL123078
and #R01GM120137, as well as funding from
Linda and Jim Lee to C.C.G. The funders had no
role in study design, data collection and analysis,
decision to publish, or preparation of the
manuscript.

**Competing interests:** The authors have declared
that no competing interests exist.

types: Lmod1 in smooth muscle, Lmod2 in cardiac muscle, and Lmod3 in skeletal muscle
[9,15–17]. Using a constitutive *Lmod2*-knockout mouse model, our laboratory demonstrated
that the absence of Lmod2 leads to rapid-onset dilated cardiomyopathy [4]; the observation
independently reached in a different study that investigated a transgenic mouse model with a
homozygous *Lmod2-piggyBac* mutation [18].

The first human mutation in *Lmod2* (*LMOD2*, p.Trp398*) was only recently discovered;
this homozygous mutation resulted in undetectable Lmod2 protein expression, extremely
short thin filaments, and severe neonatal dilated cardiomyopathy [19]. As for other leiomodin
isoforms, a child with homozygous nonsense *Lmod1* mutation died shortly after birth with a
rare condition called megacystis microcolon intestinal hypoperistalsis syndrome. Transgenic
mice with the same homozygous *Lmod1* mutation are reported to have reduced assembly of
smooth-muscle actin and impaired intestinal smooth muscle contractility before they die
shortly after birth [20]. Loss-of-function *Lmod3* human mutations present with severe congen-
ital nemaline myopathy and atrophic skeletal muscles comprised of shortened and disorga-
nized thin filaments [21–23].

In this study, to determine the consequences of longer thin filament length and how Lmod2
functions in cardiac muscle, we generated a novel transgenic mouse model (Lmod2-TG) that
overexpresses *Lmod2* in a cardiac-specific manner. Because thin filaments are significantly
shorter in the absence of Lmod2 in hearts *in vivo* [4], we hypothesized that overexpression of
Lmod2 *in vivo* would result in an increase in cardiac thin filament length. Remarkably, we dis-
covered that thin filaments lengths in intact hearts *in vivo* can be manipulated by only altering
the levels of Lmod2; Lmod2-TG mice have ~10% longer thin filaments that produce drastically
reduced active contractile force. Lmod2-TG mice also exhibit a unique combination of pheno-
types associated with cardiomyopathies, such as enlarged atrial and ventricular lumens,
increased heart mass and myofibril disarray. Furthermore, functional defects observed in
Lmod2-TG hearts extend to both systolic (reduced percent ejection fraction) and diastolic
(reduced ventricular compliance) phases of the cardiac cycle. To our knowledge, this is the
first mammalian *in vivo* model that reveals the detailed consequences of longer thin filaments,
a phenomenon that ends in cardiac failure.

## Materials and methods

Experimental samples were collected in order based on hand-markings randomly assigned at
birth. Genotype and gender information were blinded during experimental data acquisition.
Data gathered were from at least three biological replicates and at least two technical replicates.
Unless mentioned specifically, no statistically significant differences were observed between
males and females, and data reported (mean ± SEM) comes from an equal number of males
(M) and females (F) per genotype. Statistical analyses were carried out using *two-tailed t-test*
(for comparison between two samples), one-way *ANOVA* (for comparison between three or
more samples) or two-way *ANOVA* (for multiple comparisons) with recommended post-hoc
method/test (GraphPad Prism 8 for macOS; version 8.1.0 (221)). Data were considered signifi-
cant when post-hoc adjusted *P* values were less than *0.05*.

## Care and use of study animals

All animal procedures were approved by the Institutional Animal Care and Use Committee
(IACUC) at The University of Arizona. As required by our IACUC-approved animal protocol,
we used the minimum number of mice needed to obtain statistical significance (*P<0.05*)
based on power analysis. In addition, most of the physiological studies successfully carried out
in our Genetically Engineered Mouse Model (GEMM) core facility have used 6–8 animals of

any given genotype per experiment. Hence, unless stated otherwise, most of our data were generated using 6 tissue samples per genotype per experiment (multiple tissue samples can be extracted from one animal for various experimental types).

Health/status of all animals used were checked daily by laboratory animal room technician (and veterinarian, as necessary). Study animals were handled in accordance with the Guiding Principles in the Care and Use of Animals, approved by the Council of the American Physiological Society, so that they will not experience distress, pain or suffering, any more than that produced by routine handling. Briefly, before collection of experimental samples mice are deeply anesthetized (as evident by a cessation of spontaneous movement and slowing of the respiratory and heart rate) with isoflurane in an induction jar. Deeply anesthetized animals are sacrificed by either rapid decapitation (if younger than P21, age to be weaned) or cervical dislocation (P21 and older). Unused animals (or those found to be sick, injured, or otherwise unusable as determined by the laboratory animals facility's veterinarian) are euthanized by an approved painless $CO_2$ asphyxiation procedure. This procedure was chosen because it is rapid, painless and in line with the recommendations of the Panel on Euthanasia of the American Veterinary Medical Association.

## Generation and maintenance of transgenic mice with cardiac-specific leiomodin-2 overexpression

The coding sequence of full-length mouse *leiomodin-2* (NM_053098.2; 1967-bp) was amplified from cDNA generated from a neonatal mouse cardiomyocyte culture. Together with an N-terminal *c-Myc* tag following a Kozak consensus [24], *Lmod2* cDNA was cloned behind the alpha-myosin heavy chain (alpha-MHC, *Myh6*) promoter [25] and a reversed floxxed transcription termination sequence (so that exogenous *Lmod2* expression can be turned off, as needed) (S1A Fig). The transgenic colonies of two founder lines/clones were established by crossing to a Black Swiss (BLSW) strain (Taconic through JAX) and maintained by crossing *Lmod2* transgene-positive (Lmod2-TG) males with littermate *Lmod2* transgene-null (NTG) females. No significant differences in Lmod2 protein expression levels, as well as cardiac thin filament lengths were observed between the two founder lines. Presence of *Lmod2* transgene was confirmed at birth by PCR using genomic DNA. Primers used:

5'-GAGGAGGTGTGTACAGAAGATGAGGAAGAGTC (forward); and
5'-GGAGTTCCTCTGTGTTCTTCCACTGTTG (reverse).

## Generation and administration of adeno-associated virus (AAV)

Adeno-associated viruses were generated using AAV Helper-Free System (Agilent Technologies). Plasmid DNA of pHelper, pAAV-RC, pAAV-cTNT-iCre-tdTomato (Cre; a gift from Dr. William T. Pu (Boston Children's Hospital), Addgene plasmid #69916) and pAAV-cTNT-tdTomato (Tomato; control) were purified using a QIAGEN Plasmid Midi Kit. The purified DNA of Helper and AAV-RC plus either Cre or Tomato were then mixed in a 1:1:1 molar ratio. AAV-293 cells were transiently transfected with each DNA mixture (Cre or Tomato) diluted in polyethylene-imine [1 mg/mL] (Polysciences, Inc.) in a 1:4 (weight-to-volume) ratio. Twenty 15-cm culture dishes (per virus titer preparation) of AAV-293 cells at 60% confluence were used. Cells were collected 72-hour post-transfection, and lysed via four freeze-thaw cycles and an endonuclease treatment using 0.02% (vol/vol) Benzonase Nuclease (Sigma) for an hour at 37°C. Virus particles of Cre and Tomato (AAV_Cre and AAV_Tomato, respectively) were purified and concentrated by passing crude lysates through an Iodixanol density gradient (OptiPrep; Sigma) during centrifugation at 48,000 rpm for 2 hours at 18°C. Purified

virus titers were tested for vector genome-containing particles via quantitative PCR and stored at -80°C before use.

To turn transgenic expression of *Lmod2* off AAV_Cre was injected into the intrapericardial space of neonatal Lmod2-TG pups as described [26]. Briefly, approximately $2.5 \times 10^{11-12}$ genomic copies of AAV_Cre mixed in 50 μL of saline solution were injected into the intrapericardial space of postnatal day 3 (P3) Lmod2-TG pups. NTG and Lmod2-TG littermates injected with the similar amount AAV_Tomato served as experimental controls. The injected pups were left to recover on the heating pad, set at 30°C, for about 30 min before transferring back into the cage.

## Cardiac fiber mechanics (Force-Calcium relationship)

To determine the Force-Calcium relationship, as a function of diastolic sarcomere length, a modified protocol of [27] was used. The compositions of all solutions were reported previously [28]. Activating solution and relaxing solution were mixed accordingly to obtain activating solution with a free $Ca^{2+}$ range of 46.8–0.64 μmol/L. All experiments were performed at 15°C, unless otherwise mentioned.

The excised hearts from P15 and P60 mice were perfused retrograde with a modified Krebs-Henseleit solution (pH 7.4; 118.5 mmol/L NaCl, 26.4 mmol/L NaHCO$_3$, 2 mmol/L NaH$_2$PO$_4$, 15 mmol/L KCl, 1.2 mmol/L MgSO$_4$, 10 mmol/L glucose). Single unbranched fibers along the ventricular wall were then dissected and permeabilized in standard relaxing solution with 1% (vol/vol) Triton X-100 for ~4 hours at 4°C. The skinned fiber was transferred into a dish containing fresh relaxing solution (without Triton) and attached to an aluminum T-clip and mounted on hooks between a transducer (Model AE801; Sensonor) and a high-speed length controller (Model 322; Aurora Scientific). Measurements were taken at sarcomere lengths of 1.95/2.10 ± 0.03 μm using the first order diffraction band from a He-Ne laser [29].

Before acquiring Force-Calcium measurements, each fiber was stretched to a sarcomere length (SL) of 2.20–2.25 microns and a maximal activation of the fiber was performed to eliminate end-compliance from the clipping process. After the "clip-setting" activation, SL was set to 2.25 microns, and the passive tension was allowed to plateau, then a step-release was performed to obtain the passive tension measurement. This process was repeated at SL of 2.15 and 2.05 microns, as well as at 2.10 and 1.95 microns (experimental SL), to build a passive tension curve.

For the Force-Calcium curve the relaxing solution that the fiber was incubated in was briefly (3–4 min) exchanged with pre-activating solution before establishing the steady-state force using activating solution with varying free $Ca^{2+}$ concentrations (46.8–0.64 μmol/L). The fiber was then slackened (to 25% of starting fiber length) to get peak force- and zero force-level measurements. While the fiber is slack, the activating solution is replaced with relaxing solution, and then the fiber is re-stretched to the starting SL. This procedure was repeated for a total of 9 activations to build the Force-Calcium relationship curve for each individual fiber. If the final maximal activation for a particular SL was less than 85% of the initial maximal activation, the fiber was rejected from further analysis. Force-Calcium relationship curves were fit individually to a modified Hill equation as previously described [28,30]: $F_{relative} = [Ca^{2+}]^n / (EC_{50}^n + [Ca^{2+}]^n)$, where $F_{relative}$ = Force as a fraction of maximum force at saturating $[Ca^{2+}]$ ($F_{max}$), and $EC_{50} = [Ca^{2+}]$ where the $F_{relative}$ is half of $F_{max}$, and n = Hill Coefficient.

## Cardiac functional assays

**Echocardiography.** Transthoracic echocardiographs were collected on a Vevo 3100 Ultra High-Frequency Imaging Platform (FUJIFILM VisualSonics), and analyzed using AutoLV

Analysis and VevoStrain software (Vevo LAB 1.7.0). Acquisition and analysis of echocardiographs were performed according to American Society of Echocardiography guidelines. M-mode acquisitions at the papillary muscle-level were used to examine ventricular function during systole, and Pulse-wave and Tissue Doppler acquisitions were used to monitor cardiac conditions during diastole. Echocardiographs of mice at both conscious and anesthetized conditions were analyzed.

**Pressure-Volume (PV) loop.** PV-loop studies on P60 mice were conducted using a Scisense Admittance Derived Volume (ADV500) measurement system and 1.2 F catheters with 4.5 mm electrode spacing (Transonic Scisense Inc.). Mice were anesthetized and ventilated with 1–3% isoflurane using an SAR-1000 Ventilator (CWE Inc.). Body temperature was maintained at 37°C using a MouseMonitor S platform (Indus Instruments). Mice were secured and the abdomen was opened below the sternum. The apex of the left ventricle (LV) was punctured using a 28-gauge needle and the catheter was advanced into the LV. Baseline PV-loop readings were recorded during the steady state ~10 min after inserting the conductance catheter into the LV apex. The IVC was located and occluded during a sigh (pause) in ventilation to acquire load-independent indexes. Data acquisition and analysis was performed in LabScribe2 (iWorx). Pressure-Volume relationship data was analyzed using a mono-exponential fit ($P = A\,e^{\beta v + c}$) with the exponent ($\beta$) reported as the stiffness.

## Histology

Hearts were excised, dissected in half (in longitudinal and transverse sections) in PBS, fixed in 10% neutral-buffered formalin (Sigma) overnight, dehydrated, and embedded in paraffin. Ten-micron sections were stained using a Trichrome Stain (Masson) Kit (Sigma), and Picrosirius Red Stain Kit (Polysciences Inc.) according to manufacturers' protocols. Bright-field images were captured on a Zeiss Axio Imager M1 upright microscope using a Zeiss AxioCam MRc5 camera and processed linearly using ImageJ (NIH) and Photoshop CC (Adobe) software.

## Immuno-assays

**Immunoblotting.** Sections of LV free wall (10–30 mg) in a microcentrifuge tube containing 300.µL of lysis buffer (pH 7.4; 150 mmol/L NaCl, 1.5 mmol/L $MgCl_2$, 1 mmol/L EGTA, 10 mmol/L sodium pyrophosphate, 10 mmol/L sodium fluoride, 0.1 mmol/L sodium deoxycholate, 25 mmol/L HEPES, 1% Triton X-100, 1% SDS, 10% (vol/vol) glycerol, 1x protease inhibitors cocktail) were homogenized using a Bullet Blender (BBX24; Next Advance) at setting 10 for 4 min at 4°C. After spinning the crude lysate down at 16,000 x g for 15 min at 4°C, the concentration of total protein extracted (supernatant) was normalized using a Pierce BCA Protein Assay Kit (Thermo Fisher Scientific).

For soluble and insoluble fractionation experiments, all steps were carried out on ice using ice-cold buffer solutions, and all centrifugation steps were carried out at 2,000 x g for 2 min at 4°C. Briefly, fresh LV tissue sections were minced in Ringer buffer (pH 7.4; 150 mmol/L NaCl, 2 mmol/L KCl, 2 mmol/L $MgCl_2$, 10 mmol/L $KH_2PO_4$, 1 mmol/L EGTA, 0.1% Glucose), and thoroughly homogenized in Rigor buffer (pH 7.4; 100 mmol/L KCl, 2 mmol/L $MgCl_2$, 10 mmol/L $KH_2PO_4$, 1 mmol/L EDTA, 1% Triton X-100, 1x Protease inhibitors cocktail) using a tissue homogenizer (speed 3; 30-second interval up to 1.5 min). The supernatant after centrifugation was collected as a soluble fraction and the pellet was further washed four times with Rigor buffer (1x with and 3x without 1% Triton X-100). The washed pellet was then re-suspended in lysis buffer, sonicated (setting 2; 30 sec intervals up to 1 min) and spun down. Supernatant was then collected as an insoluble fraction, and concentration of extracted

proteins in both soluble and insoluble fractions were normalized using a Pierce BCA Protein Assay Kit.

Immunoblot samples were prepared in Laemmli Sample Buffer, resolved using a 4–15% gradient SDS polyacrylamide gel (Bio-Rad), and transferred onto a Protran nitrocellulose membrane (0.2-μm; GE Healthcare). The membrane was then stained with Ponceau S solution, and total density of each lane was quantified to normalize the amount of protein loaded. Before incubating in primary antibodies overnight at 4˚C, the membrane was blocked with 10% (wt/vol) milk in TBST (pH 7.4; 19 mmol/L Tris, 2.7 mmol/L KCl, 137 mmol/L NaCl, 0.1% Tween-20) for 2–4 hours at room temperature. The membrane was washed thoroughly with TBST before incubating in secondary antibodies for 2 hours at room temperature in the dark. After the immunoblots were thoroughly washed with TBST, they were scanned and analyzed using an Odyssey CLx Imaging System and Image Studio Software (Li-COR).

**Immunofluorescence and thin filament length measurements.** LV tissues were dissected into ~4.5x1.5 mm$^2$ (length x width) pieces and stretched in ice-cold PBS and fixed in 4% (vol/vol) paraformaldehyde/PBS (pH 7.4) overnight at 4˚C. Fixed tissue samples were washed with ice-cold PBS, embedded using Tissue-Tek O.C.T. Compound (Sakura Finetek), frozen and stored at -80˚C. Five-micron cryosections were cut and mounted onto #1.5 coverslips. Tissue cryosections and fixed neonatal cardiomyocytes were permeabilized in 0.2% Triton X-100/PBS for 30 min. and blocked with 2% (wt/vol) BSA/1% normal donkey serum/PBS for at least 1 hour at room temperature, and incubated in primary antibodies overnight at 4˚C. They were thoroughly washed with PBS before being incubated in secondary antibodies (and Phalloidin) for at least 2 hours at room temperature in the dark. After PBS washes, samples were mounted onto slides using ProLong Diamond Antifade Mountant (Thermo Fisher Scientific).

Fluorescent images were captured using DeltaVision RT System (GE Healthcare). Acquired images were post-processed for deconvolution using SoftWoRx software (GE Healthcare). Cardiac thin filament- and sarcomere lengths were measured using the DDecon plugin for ImageJ [31,32]. Surface areas of neonatal cardiomyocytes stained with CellMask Orange Plasma Membrane Stain (manufacturer's protocol; Thermo Fisher Scientific) were measured using ImageJ. Antibodies/fluorescent probes used included anti-leiomodin-2 (0.1–0.3 μg/mL) (E13; Santa Cruz Biotechnology), anti-tropomodulin-1 (0.1μg/mL), anti-sarcomeric alpha-actinin [1:1000] (EA-53; Sigma), Alexa Fluor Cascade Blue/405-conjugated goat-anti-mouse or rabbit, [4 μg/mL] and Alexa Fluor 488/594/Cy5-conjugated goat-anti-mouse or rabbit [2 μg/mL] antibodies. Alexa Fluor 488/594/Cy5-conjugated phalloidin [1:1000] (Invitrogen) was used to stain F-actin.

## Neonatal primary cardiomyocyte isolation and culture

Cardiomyocytes from mouse pups no older than post-natal day 3 (P3) were isolated and cultured as described [33]. Briefly, neonatal hearts (from at least 3 animals per genotype per culture) were harvested from littermate Lmod2-TG and NTG mice. Pooled hearts were then minced and digested 3x with collagenase and pancreatin for 10 min each in an incubator shaking at 230 rpm at 37˚C. Isolated neonatal cardiomyocytes were plated in a 35-mm tissue culture dish containing Matrigel-coated #1.5 coverslips at ~450,000 cells per dish and were cultured in DMEM (Gibco) with 10% (v/v) heat-inactivated FBS (HyClone) and 1% (vol/vol) penicillin/streptomycin (Cellgro). Neonatal cardiomyocytes after 4-day post-plating were fixed in 1% (wt/vol) paraformaldehyde in PBS for immunofluorescence staining (see above section).

### Real-time quantitative PCR

Total-RNA from LV tissue samples were extracted using RNeasy Fibrous Tissue Mini Kit (Qiagen), and cDNA from 500 ng/µL total-RNA was synthesized using Maxima First Strand cDNA Synthesis Kit (Thermo Fisher Scientific). Appropriate dilution of each cDNA sample was mixed with an equal volume of Maxima SYBR Green qPCR master mix (Thermo Fisher Scientific), and real-time qPCR was completed using Rotor-Gene Q cycler (Qiagen). Relative changes in gene expression were quantified using 2^(-ΔΔCT) method as described [34]. Primer sequences were obtained from qPrimerDepot (NIH), and tested to ensure their amplification efficiencies were not significantly different than that of the control gene, ornithine decarboxylase (ODC).

### Transcriptome sequencing (RNA-Seq)

Total-RNA of P7 Lmod2-TG and NTG littermates (n = 4–6 from 2–3 litters) were extracted as outlined in qPCR section and quantified using Bio-Analyzer (Agilent Technologies). Samples with RNA integrity number (RIN) greater than 7.8 and the value of 28S/18S ratio greater than 1.8 were chosen for RNA-seq analysis (n = 3 each). The rest of the transcriptome sequencing processes were completed by BGI Genomics.

### Transmission electron microscopy (TEM)

Hearts of P7 and P30 (n = 3 each per genotype per time point) were dissected in 30 mmol/L KCl (to arrest muscles in diastole), and LV tissues were further dissected into ~2x1 mm$^2$ pieces in a freshly-made fixative (pH 7.4; 2.5% glutaraldehyde, 2% paraformaldehyde, 0.1 mol/L sodium cacodylate). Samples were fixed in a fresh change of fixative overnight at 4˚C. Fixed samples were washed with 0.1 mol/L sodium cacodylate buffer (pH 7.4), and post-fixed/stained with 1% Osmium tetroxide for 1 hour on ice. The samples were then washed with 0.1 mol/L sodium acetate buffer (pH 5.2), and stained en-bloc with 2% uranyl acetate for 1 hour on ice. Stained samples were then dehydrated in graded ethanol series and embedded using Spurr Low Viscosity Embedding Kit (Sigma). Sixty-nanometer sections were cut and mounted onto uncoated copper-mesh grids and stained sequentially with 2% uranyl acetate solution in ethanol and 2% lead acetate solution. TEM images were acquired on Tecnai G2 Spirit BioTWIN microscope and processed using TEM Imaging and Analysis software (FEI). To account for potential differences in sample shrinkage, sarcomere length and Z-disc width were divided by A-band width for each sarcomere because the width of A-band (thick filament length) is consistently at ~1.6 microns. Length of a perpendicular line across the two opposing membranes of each adherens junctional structure was taken as the gap width of the adherens junctions.

## Results

### Transgenic mice with cardiac-specific *Lmod2* overexpression

We generated a novel cardiac-specific transgenic mouse line stably expressing Myc-tagged *Lmod2* under control of the alpha-MHC (*Myh6*) promoter (S1A Fig). Heterozygous *Lmod2* transgene-positive (Lmod2-TG) mice do not die prematurely and are able to breed when Lmod2-TG males are crossed with *Lmod2* transgene-null (NTG) females. However, the majority of Lmod2-TG females do not survive through pregnancy, potentially due to volume-induced cardiac stress (n = 10; percent fatality = 60%), and homozygous *Lmod2* transgenic pups do not survive past post-natal day 7 (n = 3 litters; percent fatality = 100%). Assembly of Myc-Lmod2 at thin filament pointed ends in cardiomyocytes isolated from *Lmod2* transgene-positive hearts as indicated by immunofluorescence staining with anti-Myc antibodies (*S1B*

Fig) is in agreement with the sub-cellular localization of endogenous Lmod2 protein [11]. We studied morphological and functional consequences of cardiac-specific *Lmod2* overexpression at different developmental time points: right after birth (post-natal day one, P1), when cardio-myocyte proliferation is considered complete (P7), following cardiac hypertrophy (P30), and when the mice reach sexual maturity (P60). *Lmod2* transgene-null (NTG) littermates served as controls.

## Lmod2 expression is significantly elevated in Lmod2-TG hearts

In order to measure the levels of Lmod2 protein expression, immunoblot analysis using total protein extracted from the left ventricle (LV) of Lmod2-TG and NTG littermates at all four developmental stages (see above) was performed. Lmod2 protein levels are elevated in Lmod2-TG hearts as early as P1 (2 to 3-fold; *P = 0.185*) (*Fig 1A*), and to a greater degree at P7 (9 to 10-fold, *P<0.0001*), P30 (6 to 7-fold, *P<0.0001*) and P60 (5 to 6-fold; *P = 0.003*) (Fig 1A *and* 1B). For both genotypes, no statistically significant differences in Lmod2 expression levels are observed between male and female mice at all time points examined.

To determine the levels of total Lmod2 protein assembled into myofibrils in Lmod2-TG hearts, soluble and insoluble (assembled) fractions of proteins at P7 (when the highest difference in Lmod2 expression between NTG and Lmod2-TG was observed) were collected and analyzed via immunoblot analysis (*S2A* Fig). A significant amount of Lmod2 is detected in the soluble fraction of Lmod2-TG hearts, whereas endogenous Lmod2 is found only in the insoluble (assembled) fraction in NTG hearts. Total Lmod2 levels are also significantly up in the insoluble fraction of Lmod2-TG (*S2A* Fig; ~3.3-fold; n = 4 each; *P<0.0001*), indicating that much more Lmod2 gets assembled in Lmod2-TG myofibrils, with about ~30% of exogenous Lmod2 being incorporated. To examine whether there is any compensatory response to increased *Lmod2* expression at the transcriptome-level, RNA-sequencing analyses of Lmod2-TG and NTG LV samples were performed using P7 mice. No significant difference in mRNA-levels of Lmod1, Lmod3, Tmod 1–3, as well as other thin filament proteins such as tropomyosin and troponins, between Lmod2-TG and NTG cardiac transcriptomes is detected (*S1 Table*).

## Lmod2-TG hearts have longer thin filaments as early as PD1

To determine if we did in fact manipulate (increase) the lengths of thin filaments *in vivo*, left ventricular thin filament lengths were measured. Significantly longer cardiac thin filament lengths were indeed observed in Lmod2-TG hearts at all time points measured: P1: ~3%, *P = 0.0005*; P7: ~10%, *P<0.0001*; P30: ~7%, *P<0.0001*; P60: ~5%; *P<0.0001* (Fig 1C *and* 1D; n = 140–260 measurements from 4–8 animals per genotype). The greatest difference in thin fil-ament lengths between Lmod2-TG and NTG hearts is observed at P7 (Fig 1D), when the rela-tive expression level of Lmod2 is the highest in Lmod2-TG animals (Fig 1A *and* 1B) and before the majority of phenotypic alterations are observed (see below). Remarkably, thin filament lengths in Lmod2-TG hearts at P7 were observed to be as long as 1.10-micron; this is ~ 26% longer than the average length observed in NTG hearts (0.87-micron; Figs 1C and S2B). To our knowledge these are the longest cardiac thin filaments observed in mice.

Curiously, thin filaments of Lmod2-TG animals do not grow significantly longer after P7 (Fig 1C), while those of the NTG continue to increase in length; thereby reducing the length difference between the two genotypes as the mice age. The fact that significantly longer cardiac thin filaments are observed as early as P1 and in P7 Lmod2-TG mice when there is no signifi-cant gross morphological (*S2 Table*) and functional (*S3 Table*) differences detected suggests that this phenotypic manifestation is the primary effect following cardiac-specific *Lmod2* over-expression. Alpha-MHC promoter also drives an atrial tissue expression, especially before

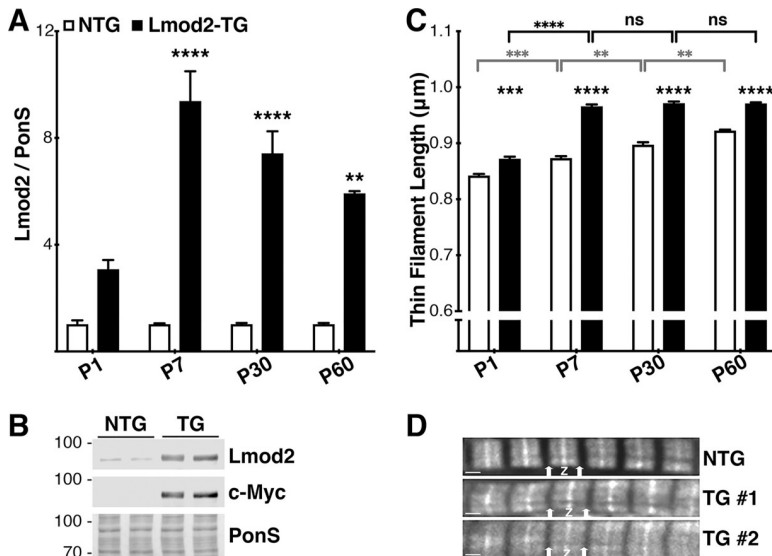

**Fig 1. High levels of Lmod2 protein expression in Lmod2-TG hearts correspond with longer thin filaments. (A)** Analysis of immunoblots of left ventricular (LV) tissue extracts of P1, P7, P30 and P60 mice probed with anti-Lmod2 antibodies. Lmod2 transgene expression is highly elevated in Lmod2-TG (*black*) when compared with endogenous Lmod2 levels in NTG (*white*) at all time points. Data are presented as the relative expression level of Lmod2 in Lmod2-TG animals when compared with NTG for each time point. (n = 4–8; Error bar = SEM; *two-tailed unpaired t-test*). **(B)** Representative immunoblot (Lmod2 and c-Myc at ~80.kDa) and Ponceau S-stained membrane (PonS) of P7 LV extracts. **(C)** Quantitative measurements of cardiac thin filament length (TFL) in P1, P7, P30 and P60 mice. TFL of Lmod2-TG hearts (*black*) are 3–10% longer when compared to those of NTG (*white*). Gray bars represent comparison of NTG TFL between each time points, and black bars represent those of Lmod2-TG (n = 4–8; Error bar = SEM; *two-way ANOVA with Tukey test*). **(D)** Representative fluorescently-conjugated phalloidin staining of LV tissue from P7 NTG and two Lmod2-TG (TG #1, #2) mice (arrows show thin filament pointed ends; Z shows Z-disc/thin filament barbed ends). Scale bar = 1 μm.

birth; therefore, we also examined atrial thin filament lengths. Interestingly, thin filaments from atrial fibers are also longer in Lmod2-TG mice, by ~5%, when compared with NTG (P15; Lmod2-TG: 0.88 ± 0.007 μm; NTG: 0.84 ± 0.014 μm; (2M, 2F) each; *P = 0.012, two-tailed unpaired t-test*). For both genotypes, no statistically significant differences in thin filament lengths are observed between male and female mice at all time points examined.

## Lmod2-TG hearts exhibit pathological hallmarks of cardiomyopathy

Although Lmod2-TG mice present with deformed hearts with enlarged atrial and ventricular chambers, a phenotypic defect that is fully penetrant by P30 (Fig 2A and 2B), morphometric

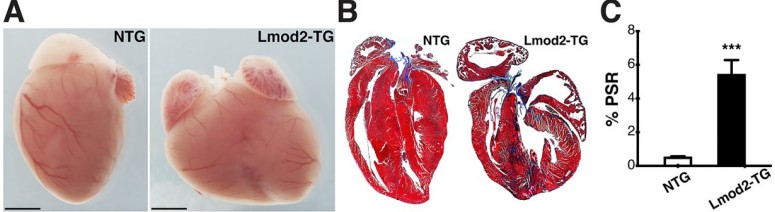

**Fig 2. All chambers of Lmod2-TG hearts are enlarged with extensive interstitial fibrosis by post-natal day 30 (P30). (A)** Atrial and ventricular chambers in Lmod2-TG hearts are larger than those in Lmod2 transgene-null (NTG) hearts. Scale bar = 1 mm. **(B)** Trichrome histological stain indicates the existence of fibrosis within the ventricular myocardium of Lmod2-TG hearts. **(C)** Quantification of Picrosirius Red-stained ventricular tissue reveals a significantly higher level of interstitial fibrosis in Lmod2-TG (*black*), when compared with NTG (*white*) (n = 6–8; Error bar = SEM; *two-tailed unpaired Welch's t-test*).

**Table 1. Primers used in this study.**

| Primer | Forward Sequence (5′ - 3′) | Reverse Complement (5′ - 3′) | Expected Size (bp) |
|---|---|---|---|
| Acta1 | CTCTCCTCAGGACGACAATC | TTTTCCATTTCCTTTCCACA | 177 |
| Myh6 | GCGCATTGAGTTCAAGAAGA | CTTCATCCATGGCCAATTCT | 103 |
| Myh7 | GTGGCTCCGAGAAAGGAAG | GAGCCTTGGATTCTCAAACG | 98 |
| Nppa | GGGGGTAGGATTGACAGGAT | AGGGCTTAGGATCTTTTGCG | 130 |
| Nppb | ACAAGATAGACCGGATCGGA | ACCCAGGCAGAGTCAGAAAC | 110 |
| ODC | ACATCCAAAGGCAAAGTTGG | AGCCTGCTGGTTTTGAGTGT | 102 |

analysis shows that heart weight-to-body weight (HW/BW), heart weight-to-tibia length (HW/TL), lung weight-to-body weight (LW/BW) and lung weight-to-tibia length (LW/TL) ratios of Lmod2-TG mice are only significantly greater than those of NTG littermate mice by P60 (*S2 Table*; $P<0.01$). Increased HW/BW and HW/TL values in Lmod2-TG mice are a typical hallmark of cardiac hypertrophy, whereas increased LW/BW and LW/TL values suggest a congested state of Lmod2-TG lungs.

While no significant morphometric differences are detected at P1 and P7, histological analysis reveals significantly increased levels of interstitial fibrosis within the ventricular walls of Lmod2-TG hearts by P30 (Fig 2B and 2C; $P<0.001$), a pathological state that contributes to ventricular stiffness, increased passive tension and decreased contractility. Reactivation of the fetal gene program is a classic indicator of cardiac disease remodeling [35]; hence, expression levels of five commonly measured fetal genes (*Acta1*, *Myh6*, *Myh7*, *Nppa* and *Nppb*; Table 1) were examined via real-time quantitative PCR (qPCR) from LV tissue. Expression levels of these genes are not significantly different in Lmod2-TG when compared to NTG at P7, although 3 out of 5 (*Acta1*, *Nppa* and *Nppb*) are trending to be higher (Fig 3, *top panels*). The expression levels of all fetal genes examined are significantly altered in Lmod2-TG hearts by P30 (Fig 3, *bottom panels*; $P<0.05$).

To examine the consequences of Lmod2 overexpression at the ultrastructural level, LV tissues of P7 and P30 Lmod2-TG and NTG littermates were subjected to transmission electron microscopy (TEM). For each sample, sarcomere length and the widths of Z-discs and A-bands were measured. Sarcomere length and Z-disc width measurements were normalized using the corresponding A-band width to adjust for sample shrinkage during the TEM procedure. No

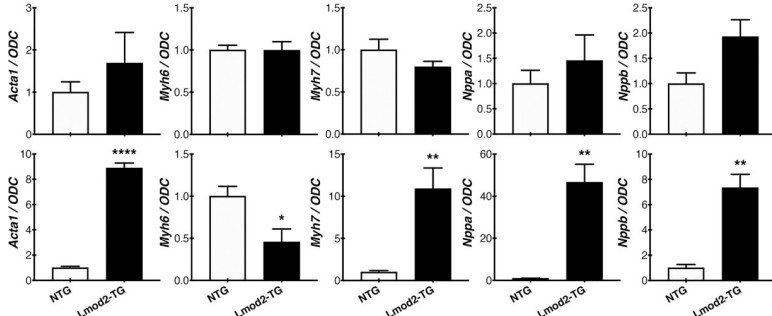

**Fig 3. Real-time quantitative PCR (qPCR) of fetal genes in NTG and Lmod2-TG shows significant disease progression at P30.** Re-activation of the fetal gene program is indicative of a diseased state of the heart; hence, qPCR of Lmod2-TG mice at P7 (*top panels*) and P30 (*bottom panels*) were examined using mRNA extracted from LV tissue. Though not statistically significant, relative expression levels of 3 out of 5 molecular heart disease markers (*Acta1*, *Nppa* and *Nppb*) are trending higher in Lmod2-TG (*black*) when compared to NTG (*white*) at P7. At P30, relative expression levels of all disease markers examined are significantly altered (n = 4–6; Error bar = SEM; *two-tailed unpaired Welch's t-test*).

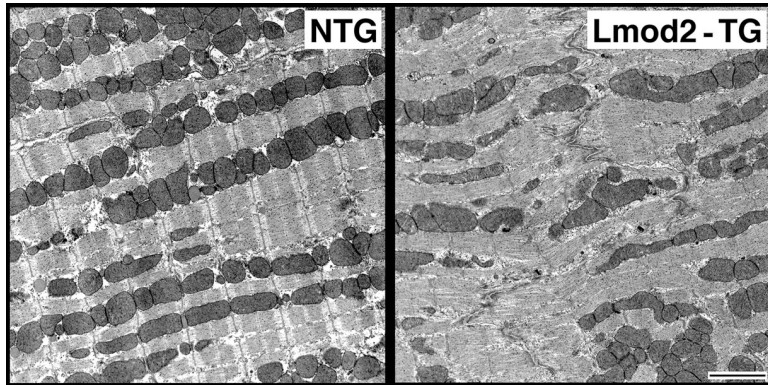

**Fig 4. Ultrastructural analysis of Lmod2-TG heart indicates extensively disrupted myofibrillar organization.**
Transmission electron micrograph of P30 LV tissue reveals disorganized myofibrillar cytoarchitecture, with out-of-register sarcomeres in Lmod2-TG hearts (representative images from n = 3 per genotype; scale bar = 2 μm).

statistically significant differences between the normalized sarcomere lengths of Lmod2-TG and NTG were observed at both P7 and P30 (*S3B* Fig; n = 3 each). However, Lmod2-TG hearts have significantly wider Z-discs when compared to NTG littermates at both P7 and P30 (*S3C* Fig; P7: ~20%, *P = 0.0001*; P30: ~28%, *P = 0.0008*). Furthermore, myofibrillar organization in Lmod2-TG heart is in complete disarray by P30 (Fig 4).

Myofibrillar disarray is often followed by dissolution of intercalated discs (ICD) [36], the junctional structures that allow transmission of electromechanical impulses between adjacent cardiac myocytes. By P30 varying degrees of perturbation of the ICDs is observed in Lmod2-TG (*S3A* Fig, *chevrons*). In particular, the gap width, a measure of the perpendicular distance between the two opposing membranes of adherens junctions, is significantly wider in Lmod2-TG hearts as early as P7 (P7: ~64%, *P = 0.0002*; P30: ~45%, *P = 0.0015*) (*S3D* Fig). Furthermore, the expression level of N-cadherin (a major adherens junction protein) is significantly upregulated in Lmod2-TG hearts (P7: ~1.4-fold; P30: ~2.7-fold; (2M, 2F) each; *P<0.05*). Altered N-cadherin expression in the myocardium has been shown to cause cardiomyopathies due to disruption of contractile dynamics by excess cadherin/catenin complexes [37–40]. Although ruptured ICDs is one prominent pathological manifestation present in cardiomyopathy [41], the abnormalities in ICDs and elevated N-cadherin expression level were first observed at P7 in Lmod2-TG, when other indicators of disease remodeling, such as gross morphology and reactivation of the fetal gene program, are yet to be significantly represented. As such, Lmod2 may have a functional role in ICD structures, independent of disease progression.

## Left ventricular Lmod2-TG fibers produce significantly less active contractile force

To investigate the mechanical consequences of longer cardiac thin filaments, the contractile properties of single unbranched trabecular fibers extracted from the left ventricular outer wall were measured. We chose to study the P60 mice first because this is the youngest age after the myosin isoform switch (from fetal *Myh6* to adult *Myh7*) has occurred. At P60 maximally activated force in Lmod2-TG was reduced by ~50% compared to NTG littermates, at both sarcomere lengths observed (1.95 and 2.10 microns; Fig 5A and 5B and *S4 Table*), along with reduction in cooperativity of myofilament proteins (Hill Coefficient, *S4 Table*). Calcium sensitivity ($EC_{50}$, *S4 Table*) was not significantly different between NTG and Lmod2-TG at

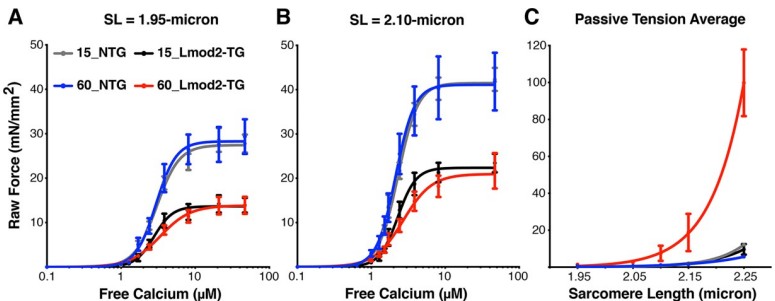

**Fig 5. Overexpression of Lmod2 negatively affects contractile force at the single myofiber level.** At P15, Lmod2-TG trabeculae fibers (*black*) show a significant drop in raw force measured at sarcomere lengths of 1.95 μm **(A)** and 2.10 μm **(B)** when compared with NTG fibers (*gray*), indicative of defective contractile force generation. Passive tension average **(C)**, as well as the calcium sensitivity ($EC_{50}$) and myofilament cooperativity (Hill coefficient) (*S4 Table*) of P15 Lmod2-TG and NTG littermates are not significantly different from each other. By P60, however, passive tension average **(C)** in Lmod2-TG (*red*) is drastically higher than those of NTG (*blue*), most likely due to extensive interstitial fibrosis. N = 1 fiber per animal from 7–9 animals. Error Bar = SEM.

1.95-micron; however, when fibers were stretched to a length of 2.10 microns the calcium sensitivity in Lmod2-TG fibers was reduced (desensitized) compared to NTG fibers.

Next, to rule out potential interference from disease progression that we believe to have already taken place in P60 Lmod2-TG mice, we analyzed mice at P15, the youngest mice we were able to get accurate measurements from. No significant differences in average passive tension, calcium sensitivity and myofilament cooperativity were observed between Lmod2-TG and NTG (Fig 5C, and *S4 Table*). However, active mechanical force produced in Lmod2-TG mice was significantly reduced, by ~47%, when compared to NTG mice. These P15 single-fiber mechanics data suggests that the primary factor contributing to reduced cardiac contractile force observed in Lmod2-TG mice is the limited interaction between thin and thick filaments.

Progressive loss of ventricular contractility can deplete circulatory reserve [42], therefore, maximal cardiac reserve available in Lmod2-TG mice was determined via LV M-mode echocardiography before and after beta-agonist treatment [43]. Dobutamine, an inotropic stimulant that targets beta1-receptors, the most common beta-adrenergic receptor type in cardiac tissue, was used in this experiment. Echocardiography data before and after dobutamine treatment (1.5 mg/kg for 5 min) of P30 Lmod2-TG and NTG littermates were analyzed. Except for an increased heart rate (~27% increase from 352 ± 13 to 448 ± 29 beats/min; (2M, 2F) each; *P<0.05*), Lmod2-TG hearts show a blunted response to dobutamine treatment, i.e., little to no significant functional improvements upon stimulation, when compared to NTG hearts that shows significant improvements in almost all criteria measured (heart rate, percent ejection fraction and percent fractional shortening shown in *S5 Table*; *P<0.05*). These data suggest that Lmod2-TG mice lack a cardiac reserve and any significant stress on their hearts may be fatal; this may explain the fatality observed during pregnancy of Lmod2-TG females.

## Lmod2-TG mice suffer from both systolic and diastolic dysfunction

To further decipher functional defects in Lmod2-TG hearts, we conducted pressure-volume loop (PV-Loop) analysis; this approach provides comprehensive load-independent measurements of ventricular systolic and diastolic function [44,45]. To maximize the reliability of the data acquired only P60 mice were utilized for this study. PV-Loop analysis of Lmod2-TG hearts indicates loss of inotropy [decreased slope in end-systolic pressure-volume relationship (ESPVR)] and loss of compliance [increased slope in end-diastolic pressure-volume

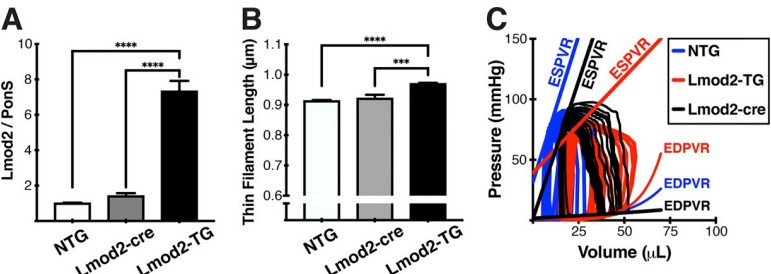

**Fig 6. Lmod2 transgene expression off: Lmod2 contributes to *in vivo* cardiac function. (A)** Analysis of immunoblots of P60 LV tissue extracts probed with anti-Lmod2 antibodies. Lmod2-cre mice, Lmod2-TG with Lmod2 transgene expression turned off, have Lmod2 protein levels similar to NTG. **(B)** Lmod2-cre mice have thin filament lengths similar to those of NTG. **(C)** Pressure-Volume Loops analysis reveals that the systolic (ESPVR, end-systolic pressure-volume relationship) and diastolic (EDPVR, end-diastolic pressure-volume relationship) dysfunctions observed in Lmod2-TG mice can be prevented by turning Lmod2 transgene expression off. N = 4–6; Error bar = SEM; non-parametric *one-way ANOVA* with *Tukey* test.

relationship (EDPVR)], indicating decreased contractility and increased stiffness in Lmod2-TG hearts (Fig 6C and S6 Table). Further, Lmod2-TG hearts present with lower peak systolic pressure with higher end-diastolic pressure, and higher relaxation constant (Tau) values, explaining the inadequate systolic ejection and insufficient diastolic relaxation functions (S6 Table). Significantly less contractile force generated in Lmod2-TG left ventricles (Fig 5A and 5B and S4 Table) is likely the major factor resulting in systolic dysfunction (i.e., primary defect) and ventricular wall stiffness, due to hypertrophy with extensive interstitial fibrosis, is likely the major contributing factor of diastolic dysfunction (i.e., secondary defect).

## Functional defects following cardiac-specific overexpression of Lmod2 are successfully prevented by turning the transgene expression off

To verify that excess Lmod2 is the primary cause of cardiac dysfunction in Lmod2-TG mice, *Lmod2* transgene expression was silenced by introducing Cre-recombinase (S1A Fig) via adeno-associated viral (AAV) transduction at P3. Strikingly, gross morphological defects, such as dilated cardiac chambers, are no longer apparent in Cre-injected Lmod2-TG mice (Lmod2-cre) ([5M, 5F], 100% absence of gross morphology compared with Lmod2-TG by P60). The levels of Lmod2 protein expressed in Lmod2-cre are comparable to NTG-level, while being significantly lower (~5-fold) than Lmod2-TG (Fig 6A; *P<0.0001*). Similarly, thin filament lengths in Lmod2-cre hearts are not significantly different from those of NTG, while being significantly shorter by ~5% than Lmod2-TG (Fig 6B; *P<0.0001*). Furthermore, cardiac functional defects observed in Lmod2-TG mice, such as systolic and diastolic performance were rescued in Lmod2-cre mice as indicated by PV-Loop analysis (Fig 6C and S6 Table). Using only one-tenth of the virus concentration (Cre_1X), which was used above to achieve ~100% prevention of disease onset as shown in Fig 6, resulted in incomplete silencing of *Lmod2* transgene expression and incomplete prevention of accompanying aberrant phenotypes (such as elongation of thin filament length and reduced percent ejection fraction; S4 Fig). Therefore, it can be concluded that excess Lmod2 protein is the primary cause that triggers thin filament elongation and cardiac dysfunction in Lmod2-TG animals.

## Discussion

Our laboratory has recently discovered a direct link between leiomodin-2 and human cardiomyopathy based on a case study of an infant with a homozygous *Lmod2* nonsense mutation

[19]. Strikingly, the clinical phenotypes observed in the patient are similar, if not identical, to the phenotypic defects observed in constitutive *Lmod2-KO* mice; in particular, extremely short thin filaments and the observation that *Lmod2* is required for post-natal survival [4,19]. In this study, we generated a novel transgenic mouse model capable of stably overexpressing *Lmod2* in cardiac tissues in a reversible manner (Lmod2-TG) to study the consequences of longer thin filaments, in the context of an intact heart.

Our data demonstrate that the maintenance of thin filament lengths is indeed critical for heart muscle function. More importantly, we show for the first time that mammalian intact hearts with longer thin filaments *in vivo* result in a complex cardiomyopathy; whereas, we previously demonstrated that shorter thin filaments *in vivo* result in rapid onset dilated cardiomyopathy. At all major postnatal developmental time points examined Lmod2 protein expression levels were elevated in Lmod2-TG mice, with the highest relative expression level difference of 9–10 fold observed at P7 (end of cardiac hyperplasia). Similarly, cardiac thin filaments were significantly longer at all stages in Lmod2-TG hearts with the largest difference in lengths observed at P7 (~10% on average with some filaments observed to be up to ~26% longer).

Interestingly, although the relative expression levels of Lmod2 are significantly elevated in Lmod2-TG at all time point examined, cardiac thin filaments in Lmod2-TG hearts do not grow significantly longer beyond P7 (average of 0.96 μm, and maximum of 1.10 μm). Meanwhile, thin filaments in NTG significantly lengthen until P60 (average of 0.92 μm, and maximum of 1.06 μm). Therefore, it can be concluded that: (1) the apparent physiological limit of mouse cardiac thin filament length is ~1.10 μm; and (2) at constant sarcomere lengths, thin filaments in Lmod2-TG overlap within individual sarcomeres, likely preventing the optimum interaction between thin (actin) and thick (myosin) filaments.

Most, if not all, of our data suggest that the primary effect of excess Lmod2 is elongated thin filaments, and that this alteration leads to cardiac disease progression. In support of this, we observe elongated thin filaments as early as P1, before any other morphological and functional alterations that we could measure were observed, such as morphology (*S5* FigB and *S2 Table*), level of interstitial fibrosis (*S5A* Fig and *S1 Table*), fetal gene expressions (Fig 3) and echocardiography (LV M-mode; *S3 Table*). Further, the greatest difference in thin filament lengths between Lmod2-TG and NTG hearts is observed at P7, when the relative expression level of Lmod2 is the highest in Lmod2-TG animals, and again before the majority of phenotypic and functional alterations are observed.

The data from our single-fiber mechanics study, especially of those acquired at P15, are consistent with the idea that the primary factor contributing to reduced cardiac contractile force observed in Lmod2-TG mice is the limited interaction between thin and thick filaments. In fact, Lmod2-TG mice at P15 have 7–8 fold higher Lmod2 expression and ~8% longer thin filaments that produce ~47% less active contractile force. This 8% increase in thin filament length in hearts of Lmod2-TG is predicted to result in a rightward shift in the sarcomere length-tension relationship, and consequently, a 10–20% reduction in maximum tension at sarcomere lengths on the ascending limb of the relationship; the reduction predicted to be due to interference of overlapping thin filaments [29,46–48]. Therefore, the ~47% reduction in maximum force observed in Lmod2-TG animals at both sarcomere length measured cannot be solely explained by an increase in thin filament lengths because the longer sarcomere length tested (2.10-micron) is outside the double overlap suggested by an 8% increase in thin filament lengths. As such, some other mechanism, in addition to elongated thin filaments, must be contributing to the reduction in production and/or transmission of contractile force within the isolated single fibers.

We hereby propose two additional ways that excess Lmod2 can contribute to impaired production of contractile force. First, excess soluble Lmod2 may bind to the sides of actin (thin)

filaments and effectively limit the interaction of thin and thick filaments. This idea is based on reports that excess Lmod2 in isolated cardiomyocytes can assemble along the length of the thin filaments and interfere with proper myosin cross-bridge formation leading to a decrease in force production [11,49]. Second, our TEM data show that Lmod2-TG hearts have disorganized myofibrils, wider Z-discs and ruptured intercalated discs. We thus propose that longer thin filaments in Lmod2-TG heart leads to myofibrillar disarray that reduces sarcomeric tension, which in turn disrupts assembly and maintenance of intercalated disc (ICD) structures. Because ICD structures between adjacent cardiomyocytes ensure efficient propagation of electromechanical impulses, disrupted ICDs observed in Lmod2-TG hearts would be predicted to interfere with mechanical force transmission from one cardiomyocyte to another, contributing to the total contractile force deficit.

When Lmod2 transgene expression is turned off in Lmod2-TG mice via adeno-associated viral transduction of Cre-recombinase, all detectable phenotypical and functional aberrations observed in Lmod2-TG mice, such as longer thin filament length, enlarged cardiac chambers, and systolic and diastolic dysfunction are successfully prevented. We observed that the degree of phenotypical and functional defects prevented in Lmod2-cre hearts are dependent on the amount of Cre-recombinase injected into Lmod2-TG pups. These data suggest that cardiac-specific overexpression of Lmod2 with subsequent thin filament elongation is the primary causative factor leading to eventual heart failure in Lmod2-TG mice.

To our knowledge, Lmod2 is the only protein shown to have a direct correlation between the amount of expressed protein, thin filament length and cardiac function *in vivo*. Specifically, the absence of Lmod2 in cardiac-specific *Lmod2*-KO mice results in shorter thin filaments with the lengths linked to the amount of knockdown via tamoxifen dose [50], while here we show that increasing the levels of Lmod2 expressed in cardiac muscle is positively correlated with increased thin filament lengths. Although this result is what we predicted from assays in cell culture, it is remarkable that we can manipulate thin filaments lengths so precisely in intact hearts because of the presence of known and presumably unknown regulatory mechanisms. We conclude that Lmod2 dictates *in vivo* cardiac function by its contribution to sarcomeric integrity and actin-myosin interactions via regulation of thin filament length, and that the Lmod2-TG model serves as the model where we can learn how repeated cellular-level insults (functional performance of elongated thin filaments at each contraction-relaxation cycle) progress into heart failure.

## Supporting information

**S1 Checklist. *PLOS ONE* humane endpoints checklist.**
(DOCX)

**S1 Fig. Schematic of experimental design used to generate the Lmod2-TG transgenic mouse strain.**
(DOCX)

**S2 Fig. Higher levels of Lmod2 expression are detected in both soluble and insoluble fractions of Lmod2-TG hearts, which have consistently longer thin filaments.**
(DOCX)

**S3 Fig. Quantitative analysis of TEM indicates that Lmod2-TG hearts possess significantly wider Z-discs and adherens junctions, hallmarks of cardiac myopathy.**
(DOCX)

**S4 Fig. Incomplete silencing of *Lmod2* transgene expression results in partial prevention (rescue) of aberrant phenotypes in Lmod2-TG animals.**
(DOCX)

**S5 Fig. The levels of interstitial fibrosis in Lmod2-TG hearts in younger mice, and cardiomyocyte size in Lmod2-TG neonatal cardiomyocytes are not significantly different from those of NTG.**
(DOCX)

**S1 Table. Comparison of RNA transcript levels between NTG and Lmod2-TG.**
(DOCX)

**S2 Table. Morphometric analyses of NTG and Lmod2-TG.**
(DOCX)

**S3 Table. Left ventricular (LV) echocardiography analyses of NTG and Lmod2-TG mice via M-mode.**
(DOCX)

**S4 Table. Summary of single cardiac fiber mechanics study.**
(DOCX)

**S5 Table. Cardiac reserve in NTG and Lmod2-TG mice.**
(DOCX)

**S6 Table. Pressure-Volume Loop (PV-Loop) analysis.**
(DOCX)

## Acknowledgments

We would like to thank Max Zelikovsky, David O'Neil Lyons and Greg Lyons for their assistance with mouse colony maintenance and genotyping; Dr. William Day of the University of Arizona TEM core for his help with the transmission electron microscopy procedures; and Drs. Tom Doetschman and Teodora Georgieva of the University of Arizona GEMM Core for generating the cardiac-specific Lmod2 overexpression mouse.

## Author Contributions

**Conceptualization:** Lei Mi-Mi, Carol C. Gregorio.

**Data curation:** Lei Mi-Mi, Gerrie P. Farman, Joshua Strom, Christopher T. Pappas, Carol C. Gregorio.

**Formal analysis:** Lei Mi-Mi, Gerrie P. Farman, Rachel M. Mayfield, Joshua Strom.

**Funding acquisition:** Lei Mi-Mi, Carol C. Gregorio.

**Investigation:** Lei Mi-Mi, Gerrie P. Farman, Rachel M. Mayfield, Joshua Strom.

**Methodology:** Lei Mi-Mi, Gerrie P. Farman, Rachel M. Mayfield, Joshua Strom, Miensheng Chu, Christopher T. Pappas, Carol C. Gregorio.

**Project administration:** Lei Mi-Mi, Carol C. Gregorio.

**Resources:** Gerrie P. Farman, Miensheng Chu, Christopher T. Pappas, Carol C. Gregorio.

**Supervision:** Lei Mi-Mi, Christopher T. Pappas, Carol C. Gregorio.

**Validation:** Lei Mi-Mi, Gerrie P. Farman, Miensheng Chu.

**Visualization:** Lei Mi-Mi.

**Writing – original draft:** Lei Mi-Mi.

**Writing – review & editing:** Lei Mi-Mi, Gerrie P. Farman, Joshua Strom, Christopher T. Pappas, Carol C. Gregorio.

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
