## [Decision Letter · Decision Letter 0]

1 Oct 2019

PONE-D-19-23641

In vivo elongation of thin filaments results in heart failure

PLOS ONE

Dear Dr. Gregorio,

Thank you for submitting your manuscript to PLOS ONE. After careful consideration, we feel that it has merit but does not fully meet PLOS ONE’s publication criteria as it currently stands. Therefore, we invite you to submit a revised version of the manuscript that addresses the points raised during the review process.

ACADEMIC EDITOR: The authors should better highlight limitations of the study. Moreover, it is important to measure capillary density, cardiomyocyte size and collagen content.

We would appreciate receiving your revised manuscript by Nov 15 2019 11:59PM. To enhance the reproducibility of your results, we recommend that if applicable you deposit your laboratory protocols in protocols.io, where a protocol can be assigned its own identifier (DOI) such that it can be cited independently in the future. For instructions see: http://journals.plos.org/plosone/s/submission-guidelines#loc-laboratory-protocols

We look forward to receiving your revised manuscript.

Kind regards,

Vincenzo Lionetti, M.D., PhD

Academic Editor

PLOS ONE

**Journal Requirements:**

**Comments to the Author**

1. Is the manuscript technically sound, and do the data support the conclusions?

Reviewer #1: Yes

Reviewer #2: Yes

2. Has the statistical analysis been performed appropriately and rigorously? 

Reviewer #1: Yes

Reviewer #2: Yes

3. Have the authors made all data underlying the findings in their manuscript fully available?

Reviewer #1: Yes

Reviewer #2: Yes

4. Is the manuscript presented in an intelligible fashion and written in standard English?

Reviewer #1: Yes

Reviewer #2: Yes

5. Review Comments to the Author

Reviewer #1: The Authors have developed a novel transgenic mouse model stably overexpressing Lmod2 in cardiac tissues in a reversible manner, in order to study the consequences of longer thin filaments. This sudy confirms that the maintenance of thin filament lengths is critical for heart muscle function and shows that mammalian intact hearts with longer thin filaments in vivo result in a complex cardiomyopathy. The paper is well written, the methods appear appropriate and the results are clearly stated and quite interesting.

Reviewer #2: Review of PONE-D-19-23641 submitted by Mi-Mi et al.

Title of MS: In vivo elongation o thin filaments results in heart failure.

This is a beautifully written, well documented and carefully deduced conclusions on the effects of increasing the amount of a thin filament pointed end associated protein leiomodin-2 in mouse models. Increasing the level of this protein resulted in a 10 to sometimes 26% longer thin filaments. Examination of the effects of raising the intracellular levels of lmod2 also lead to effects on the Z-Bands and intercalated discs in the mouse hearts.. The sarcomeres from muscle expressing this higher level of Lmod2 resulted in a loss of upto 50% loss of contractile forces resulting in cardiac failure. It is very impressive that their experiments in which they lower the high levels on Lmod2 lead to a reversal of the effects of the high levels of Lmod2. The authors have previously demonstrated that lowering the amount of Lmod in the cardiomyocytes lead to shorter filaments, and also cardiac defects. The combination of these two reports demonstrates the importance of Lmod in the health of the heart. This manuscript is a first-rate contribution to our understanding of the role of Lmod2 in myofibril assembly, and how changing the normal levels of Lmod2 can lead to cardiac failure. This manuscript should be published without changes.

6. PLOS authors have the option to publish the peer review history of their article (what does this mean?). If published, this will include your full peer review and any attached files.

Reviewer #1: No

Reviewer #2: No

---

## [Author Response · Author response to Decision Letter 0]

8 Nov 2019

From Academic Editor (1 of 2): “The authors should better highlight limitations of the study.”

 The only limitation of this study that we are aware of is that we are not able to study the mice homozygous for the Lmod2 transgene. This is addressed as follows:

Heterozygous Lmod2 transgene-positive (Lmod2-TG) mice do not die prematurely and are able to breed when Lmod2-TG males are crossed with Lmod2 transgene-null (NTG) females. However, the majority of Lmod2-TG females do not survive through pregnancy, potentially due to volume-induced cardiac stress (n = 10; percent fatality = 60%), and homozygous Lmod2 transgenic pups do not survive past post-natal day 7 (n = 3 litters; percent fatality = 100%).

Thus, analysis of homozygous Lmod2 transgenic mice could not be included in this study. See page16, lines 313-318.

From Academic Editor (1 of 2): “Moreover, it is important to measure capillary density, cardiomyocyte size and collagen content.”

 We are a bit confused about the question the Editor would like us to address with this request but, hopefully, we sufficiently addressed the request below. In short, we do not have any data that demonstrates statistically significant differences between Lmod2-TG and NTG mice, in terms of capillary density, cardiomyocyte size or collagen content before Lmod2-TG hearts display cardiac disease remodeling (P30; Fig. 2). In comparison, cardiac thin filament lengths are already 3% longer at P1, and contractile force reduced by 47% at P15.

- RE: Capillary density

 We deduce that this query arose from two possible scenarios: (1) increases in capillary density accompany hypertrophied hearts; or (2) decreases in capillary density give way to tissue hypoxia, cell death and interstitial fibrosis. Both scenarios could lead to contractile dysfunction and heart failure.

 Note, #1 is unlikely a primary factor contributing to heart failure because we only observe cardiac hypertrophy (i.e., increased heart mass) in Lmod2-TG mice at P60 (Supporting Table S3), when disease remodeling has already taken place.

 As for #2, we found that: (i) levels of interstitial fibrosis in Lmod2-TG are only significantly elevated after P30 (Fig. 2C), and not at P1 and P7 as indicated by Picrosirius Red-staining and quantification (%PSR) (which is now added as a Supporting Figure S5A; n = 4-10; error bars = SEM); and (ii) passive tension average (a criterion linked to fibrosis levels) between NTG and Lmod2-TG is not statistically significant at P15 (Fig. 5C).

 It is also important to note that we do not detect changes in the levels of Lmod1, the leiomodin isoform predominantly expressed in smooth muscles, as well as other markers of pathological capillary densities, in our Lmod2-TG model (Supporting Table S1). Together, these data suggest that changes in capillary density are unlikely a primary contributing factor to heart failure in the Lmod2-TG mice.

Added: Supporting Figure S5A

Updated: Supporting Table S1

- RE: Cardiomyocyte size

 Surface areas of neonatal cardiomyocytes extracted from P2 Lmod2-TG and NTG littermates at day 1 (D1), day 3 (D3) and day 5 (D5) post-plating were measured using ImageJ. No statistically significant difference was detected at all time points examined. These data are now added as a Supporting Figure S5B (n = 30 each; error bars = SEM).

Added: Supporting Figure S5B

- RE: Collagen content

 We used Picrosirius-Red staining to detect collagenous deposits (see above). No significant difference between NTG and Lmod2-TG in the percent Picrosirius Red-stained (%PSR) P1 and P7 LV sections was detected (n = 4-10). In addition, we detected no significant difference in collagen gene expressions in our RNA-Seq between P7 NTG and Lmod2-TG. These data have been added to Supporting Table S1 (P < 0.05; n = 3 per genotype).

Updated: Supporting Table S1

---

## [Decision Letter · Decision Letter 1]

21 Nov 2019

In vivo elongation of thin filaments results in heart failure

PONE-D-19-23641R1

Dear Dr. Gregorio,

We are pleased to inform you that your manuscript has been judged scientifically suitable for publication and will be formally accepted for publication once it complies with all outstanding technical requirements.

With kind regards,

Vincenzo Lionetti, M.D., PhD

Academic Editor

PLOS ONE

Additional Editor Comments (optional):

Reviewers' comments:

Reviewer's Responses to Questions

**Comments to the Author**

1. If the authors have adequately addressed your comments raised in a previous round of review and you feel that this manuscript is now acceptable for publication, you may indicate that here to bypass the “Comments to the Author” section, enter your conflict of interest statement in the “Confidential to Editor” section, and submit your "Accept" recommendation.

Reviewer #1: All comments have been addressed

2. Is the manuscript technically sound, and do the data support the conclusions?

Reviewer #1: Yes

3. Has the statistical analysis been performed appropriately and rigorously? 

Reviewer #1: Yes

4. Have the authors made all data underlying the findings in their manuscript fully available?

Reviewer #1: Yes

5. Is the manuscript presented in an intelligible fashion and written in standard English?

Reviewer #1: Yes

6. Review Comments to the Author

Reviewer #1: This paper is well written and the study has been properly conducted. The data as to the role of Lmod2 in cardiac failure are novel and of interest.

7. PLOS authors have the option to publish the peer review history of their article (what does this mean?). If published, this will include your full peer review and any attached files.

Reviewer #1: No

---

## [Editor Report · Acceptance letter]

27 Nov 2019

PONE-D-19-23641R1 

*In vivo* elongation of thin filaments results in heart failure 

Dear Dr. Gregorio:

I am pleased to inform you that your manuscript has been deemed suitable for publication in PLOS ONE. Congratulations! Your manuscript is now with our production department. 

With kind regards,

on behalf of

Prof. Vincenzo Lionetti 

Academic Editor

PLOS ONE